# Cashew Tree Pollen: An Unknown Source of IgE-Reactive Molecules

**DOI:** 10.3390/ijms20102397

**Published:** 2019-05-15

**Authors:** Daniele Danella Figo, Karine De Amicis, Denise Neiva Santos de Aquino, Fabiane Pomiecinski, Gabriele Gadermaier, Peter Briza, Clovis Eduardo Santos Galvão, Jônatas Bussador do Amaral, Carlo de Oliveira Martins, Fabio Fernandes Morato Castro, Jorge Kalil, Keity Souza Santos

**Affiliations:** 1Disciplina de Imunologia Clínica e Alergia, Faculdade de Medicina da Universidade de Sao Paulo, São Paulo 01246-903, Brazil; danidanella@hotmail.com (D.D.F.); karineamicis@gmail.com (K.D.A.); fabio.castro@imabrasil.com.br (F.F.M.C.); jkalil@usp.br (J.K.); 2UNIFOR, Universidade de Fortaleza, Fortaleza 60811-905, Brazil; deniseneiva@gmail.com (D.N.S.d.A.); fabiane.alergia@hotmail.com (F.P.); 3Department of Biosciences, Paris-Lodron-University of Salzburg, 5020 Salzburg, Austria; Gabriele.Gadermaier@sbg.ac.at (G.G.); Peter.Briza@sbg.ac.at (P.B.); 4Serviço de Imunologia Clinica e Alergia, Hospital das Clinicas HCFMUSP, Faculdade de Medicina, Universidade de Sao Paulo, Sao Paulo 01246-903, Brazil; clovis.galvao@hc.fm.usp.br; 5ENT Research Lab, Department of Otorhinolaryngology—Head and Neck Surgery, Federal University of Sao Paulo, Sao Paulo 04021-001, Brazil; amaraljb@gmail.com; 6Department of Biological Chemistry, John Innes Centre, Norwich NR4 7UH, UK; carloleia@gmail.com; 7Disciplina de Imunologia Clinica e Alergia, Hospital das Clinicas HCFMUSP, Faculdade de Medicina, Universidade de Sao Paulo, Sao Paulo 01246-903, Brazil; 8Laboratorio de Imunologia, Instituto do Coracao, Hospital das Clinicas da Faculdade de Medicina da Universidade de Sao Paulo, Sao Paulo 01246-903, Brazil; 9Institute for Investigation in Immunology (iii), INCT, Sao Paulo 01246-903, Brazil

**Keywords:** novel allergens, Brazil, pollinosis, proteome, shotgun analysis, aeroallergens

## Abstract

Pollinosis is sub-diagnosed and rarely studied in tropical countries. Cashew tree pollen has been reported as an allergen source although the knowledge of its immunoglobulin E (IgE)-reactive molecules is lacking. Therefore, this work aimed to identify IgE-reactive molecules and provide a proteomic profile of this pollen. From the 830 proteins identified by shotgun analysis, 163 were annotated to gene ontology, and a list of 39 proteins filtered for high confidence was submitted to the Allfam database where nine were assigned to allergenic families. Thus, 12 patients from the northeast of Brazil with persistent allergic rhinitis and aggravation of symptoms during cashew flowering season were selected. Using a 2D-based approach, we identified 20 IgE-reactive proteins, four already recognized as allergens, including a homolog of the birch isoflavone-reductase (Bet v 6). IgE-reactivity against the extract in native form was confirmed for five patients in ELISA, with three being positive for Bet v 6. Herein, we present a group of patients with rhinitis exposed to cashew tree pollen with the first description of IgE-binding proteins and a proteomic profile of the whole pollen. Cashew tree pollen is considered an important trigger of rhinitis symptoms in clinical practice in the northeast of Brazil, and the elucidation of its allergenic molecules can improve the diagnostics and treatment for allergic patients.

## 1. Introduction

Allergic rhinitis (AR), as defined by the Allergic Rhinitis and its Impact on Asthma (ARIA) guidelines, is a well-defined endotype. It is an inflammatory condition caused by an immunoglobulin E (IgE)-mediated response to a spectrum of allergen sources including pollens, dust mites, cockroaches, animal dander, and molds [1]. The clinical manifestation of pollen sensitization is frequently termed pollinosis, which is a seasonal allergic disease typically recurring at the same time of the year. Clinically, it is frequently characterized by rhinoconjunctivitis and/or asthma. Patients manifest ocular pruritus with conjunctival hyperemia, runny or itchy nose, and sneezing that can include nasal obstruction. Bronchial hyperreactivity can occur in 15%–20% of patients [2].

A high incidence of pollinosis is found in the temperate climate regions of Europe, Asia, and the United States of America (US). In the US, it is estimated that 8.2% of the population is annually affected [3], while in Europe, the prevalence of sensitization to pollen allergens is estimated to be higher than 40% [4]. Although the first article on pollinosis in Brazil was published by Carini et al. in 1908 suggesting the possibility of the appearance of the “pollen disease” [5], for many years, pollinosis was considered non-existent in the country, with only scarce literature on the topic.

In 1987, Rosário-Filho et al. published a study carried out with 50 rhinitis patients from the south of Brazil who presented an exacerbation of symptoms during spring season [6]. *Lolium multiflorum*, known as ryegrass, is since then considered the main cause of pollinosis in Southern Brazil [7].

Currently, the diagnosis of AR is based on detailed clinical history and further IgE-specific tests that should be performed according to patient exposure. The in vivo or in vitro determination of specific IgE can confirm diagnosis and also guide therapy. Allergen immunotherapy (AIT) is the only curative approach for allergic diseases, targeting their antigen-specific immunologic mechanisms [8]. In most cases, AIT is performed with a total extract of the causative allergen source, but some molecules are already under clinical trials for this purpose [9]. Both the in vitro IgE assays based on allergenic extracts and those based on molecules will probably coexist in clinical practice for many years to come [10].

The cashew tree (*Anacardium occidentale* L.) is cultivated in the tropics between 25° N and 25° S and is supremely adapted to hot lowland areas with a pronounced dry season, where mango and tamarind trees also thrive [11]. In 2017, the global production of cashew nuts (as kernel) was 3,971,046 tons, led by Vietnam, India, and Côte d’Ivoire with 22%, 19%, and 18% of the world’s total respectively. Benin, Guinea–Bissau, Tanzania, Mozambique, Indonesia, and Brazil also significantly produce cashew kernels. This tree is native to Brazil and dispersed in almost all its territory. Fortaleza, in Ceará state, is one of the main producers of cashew nuts in the country [12]. Cashew trees belong to Sapindales, a taxonomic order that has never been reported to contain inhaled allergens, although food allergens have been described to be present in pineapple, lemon, sweet orange, tangerine, lychee, and pistachio that belong to the same order. The presence of three allergens have been identified in cashew nuts—Ana o 1, Ana o 2 and Ana o 3 [13]—and Ana o 1 and 2 have been detected in cashew juice [14]; however, no description exists of IgE-reactive proteins in cashew tree pollen.

In Brazil, pollinosis is mainly considered to occur in latitudes greater than 25° S, with altitudes above 400 meters, in inner continental areas under environmental conditions similar to Europe and the Northern hemisphere [3]. So far, no studies have been conducted in other parts of the country particularly under tropical climate conditions. Notwithstanding, we describe here a group of subjects from the northeast of Brazil presenting exacerbated allergic symptoms during the cashew tree flowering season and identify IgE-reactive molecules as potential triggers of symptoms.

## 2. Results

Cashew particles showed strong auto fluorescence [15], and the employment of the primary fluorescence of pollen [16] eases their isolation from the other components of an air sample. Images obtained in phase contrast microscopy showed the anthers extracted from flowers contained considerable amounts of pollen (Figure 1) with a median size of 31.158 ± 3.073 µm (considering 15 pollen particles), analogous to the diameter of known allergenic pollens.

Phenol extraction followed by ammonium acetate precipitation in methanol produced the best protein yield, resulting in 1 mg of protein obtained from 60 mg of pollen sacs.

Trypsin digestion of the total extract allowed the identification of 830 individual proteins by shotgun analysis. A list with 39 distinct proteins was obtained after filtering for high confidence. From those, nine proteins were assigned to an already described allergenic protein family according to AllFam (Figure 2a and Table 1).

In addition to the information about the presence of members of known allergenic protein families, the results on identified proteins of cashew tree pollen were obtained. After data refinement, where unknown products and redundant proteins were excluded, a total of 163 unique proteins were identified and annotated to gene ontology. Detailed information of gene ontology annotation regarding biological processes, cellular components, molecular functions, and protein class are shown in Figure 2c.

Patients included in our study presented typical symptoms of allergic rhinitis, conjunctivitis, and asthma, and two presented dermatitis. Prick tests were negative for all patients for both grass pollen extracts tested, and some showed sensitization to mites *Dermatophagoides farinae*, *Dermatophagoides pteronyssinus,* and *Blomia tropicalis* besides mold (Table 2). Notably, none of them presented clinical symptoms of allergy after the ingestion of cashew nuts or fruits.

The primary investigation of IgE reactivity by 1D Western blotting (1D WB) using individual serum revealed different IgE sensitization profiles with several bands being recognized mainly from 30 to 100 kDa (Figure 3a).

From a broad 2D Western blotting (2D WB) (isoelectric point 3–10) with pooled sera, 28 spots were assigned between 22 and 75 kDa and isoelectric points from 4 to 10 and, therefore, excised from gel for identification (Figure 2b). From those, 12 presented positive hits upon the database search. Seven were described to belong to allergenic protein families. Two of those have been previously described as allergenic proteins in other pollens, i.e., isoflavone reductase [17] (spot 19) and β-1,3-glucanase [18] (spot 22) (Table 3).

Since most proteins in the isoelectric point (pI) 3–10 gel were concentrated in the pI 4–7 region, another gel with a narrower pI was run for further analysis. From this gel, 29 spots were excised and identified, distributed mainly within 40–90 kDa, pI 4.5–6.5. Two of them have been previously identified as allergens, i.e., heat shock protein 70 kDa [19] (spots 30, 32, 50, 52) and fructose-bisphosphate aldolase (spot 47) (Figure 3b).

In total, 40 spots from both gels were identified, and after filtering by −logp > 15, the exclusion of the uncharacterized and the selection of those with >2 unique peptides resulted in a list with 32 identified spots, corresponding to 20 distinct proteins assigned to seven allergenic families (Table 3). According to our search in Allergome and the World Health Organization/International Union of Immunological Societies (WHO/IUIS) Allergen nomenclature sub-committee database, the other identified proteins in 2D WB are so far not reported to be allergenic.

Considering that the cashew tree may present IgE cross-reactivity with tree pollens having taxonomical similarities, all plant species identified in the database search for homolog IgE-reactive proteins were grouped in a phylogenetic tree to check for the proximity of species (Figure 3c). The closest species to *Anacardium occidentale*, which also belong to the order Sapindales, were *Mangifera indica* and *Citrus sinensis*, from which four homologous proteins were identified: β-glucosidase 44-like (*C. sinensis*), sucrose synthase, β-1,3-glucanase, and β-galactosidase (*M. indica*), the last two having been reported as allergens [18,20,21]. Five other species present inhaled allergens already described, *N. tabacum*, *Helianthus annuus*, *Glycine soja*, *Glycine max,* and *Corylus avellana*, but those are taxonomically more distant.

IgE reactivity against the whole pollen extract in native form was confirmed for five out of 12 patients in ELISA (Figure 4a). An isoflavone reductase, like birch isoflavone-reductase (Bet v 6), was identified at spot 19, and in shotgun analysis, therefore the sera from five positive patients for the whole extract and also five non-sensitized sera were tested for rBet v 6. From those, three out of five patients were positive while nonallergic were negative (Figure 4b).

## 3. Discussion

According to Driessen et al. [22], pollens that cause allergies are between 10 and 40 microns in size with most being between 20 and 35 microns. Using fluorescence microscopy, we verified that the cashew tree pollen diameter is around 30 µm. Fluorescence is a useful index to distinguish between biological and non-biological airborne particles [23]; thus, fluorescence microscopy was applied as a suitable technique for the investigation of pollen particles [24,25,26].

The allergy to cashew tree pollen is believed to occur in the northeast of Brazil, but no detailed scientific reports are available. The ability of cashew tree pollen to trigger asthmatic responses in allergic patients has been reported by Fernandes and Mesquita et al. in 1995 [27] based on skin tests and bronchial provocation but without the description of proteins involved. In 2002, Menezes et al. [28] tested 80 Brazilian asthmatic patients sensitized to different pollens, and all of them were positive in intradermal tests with a cashew tree pollen extract. The group of patients enrolled in this study presented with rhinitis with sensitivity to mites and was negative for the mix pollens grasses tested. These negative results for grass pollens is probably because species present on these extracts are not common in tropical regions like Brazil.

Within this study, we present the first report of IgE-reactive molecules for cashew tree pollen. The first approach by 1D Western blotting showed different patterns of IgE recognition among the investigated patients, including the presence of strongly reactive protein bands. Some patients present several IgE-reactive bands, while others show a sparser profile. These several proteins recognized by allergic patients that are not recognized by non-atopic individuals can represent novel allergens. Reactive molecules present in non-atopic individuals can be solely IgE-sensitive or IgE cross-sensitive without clinical expression. Lastly, there is a protein band at approximately 31 kDa recognized by all patients and controls that probably results from unspecific binding by the second antibody.

In this study, we provided for the first time a proteomic shotgun analysis of cashew tree pollen. The biological processes of the identified proteins in cashew tree pollen were annotated mainly to metabolic and cellular processes and stimuli responses, which is in accordance with other pollen proteomic studies [29]. The resulting proteomic profile demonstrated the distribution of proteins in cellular components as well as their molecular function and biological processes that can be helpful for further studies of this plant.

Although phylogenetic tree did not evidence a close relationship between the cashew tree and other plants with described aeroallergens, a possibility of this occurrence can be the lack of information available at protein databanks for these species.

Some identified proteins found in the shotgun analysis belong to allergenic protein families present in inhaled allergen sources. Calmodulin belongs to the Elongation Factor hand family, which is a large allergenic family that includes Bet v 4, Che a 3, Cyp c 1, and Phl p 7; Cu/Zn superoxide dismutase (Ole e 5 allergen) belongs to a family with the same name.

Other allergenic families that include pollen allergens were also identified by 2D WB, reinforcing the reliability of data. The isoflavone reductase family includes pollen allergens such as Bet v 6, from *Betula verrucosa* [30] and Cor a 6 [31], from hazelnut trees. These proteins have also been found in apple, pear, orange, mango, lychee, carrot, banana, pea, and chickpea. Bet v 6 has a sequence identity of 56%–80% compared to homologous proteins from various other plants [30]. The β-1,3-glucanase and X8 domain families consist of pollen allergens such as Ole e 4, Ole e 9, Ole e 10 and contain a β-galactosidase described as IgE-reactive for palm tree pollen [20,21]. Ole e 9, a β-1,3-glucanase, is a major olive tree allergen involved in 65% of the allergic responses of patients suffering from allergy to olive pollen [18]. As a panallergen, this protein is present in different allergen sources, other fruits, and vegetables such as kiwi, banana, and avocado [32]. Besides, it has been shown to be a relevant allergen in the pollen–latex–fruit syndrome. Lastly, the heat shock protein 70 kDa family comprises an allergenic protein found in hazelnut pollen known as Cor a 10 [19].

A group of proteins was identified as IgE-reactive in 2D WB that although not yet recognized as pollen allergens was assigned to the allergenic families Prolamin, GDSL-hydrolase, Profilin, and fructose-bisphosphate aldolase class I, therefore, being potential novel allergens from this source. Consistent with the presence of several IgE-reactive bands, ELISA showed that five patients (1, 5–8) were positive for the native proteins present in the whole extract representing 41.7% of patients (5/12). Positive patients’ sera were also tested for Bet v 6 and out of those five, three (1, 6, and 7) were also positive for Bet v 6 (3/5) representing 60% of positive in the whole extract and 25% of the total group (3/12). These results are compatible with 1D WB where one can see that positive patients react to a protein band around 35 kDa, which might correspond to Bet v 6, while for the other two negative patients (5 and 8), no recognition in this same region was found.

Patient 6 presented a relative disagreement (quantitative differences) between allergen extract and allergen molecular IgE assay results, while patient 5 presented an absolute disagreement between them. Patient 5 is positive for allergen extract but is negative for Bet v 6. The IgE antibodies of this serum might recognize one/several particular molecules in the extract and its genuine species-specific (major) components of the extract and not minor, highly cross-reactive components, such as Bet v 6. Patient 6 presented higher levels for Bet v 6 and lower levels for pollen extract, indicating Bet v 6 is of low abundance in the extract [33].

ELISA revealed that only five patients were positive for the native proteins present in the whole extract, but on the other hand, all 12 included patients were positive in WB recognizing several protein bands. It is not entirely clear why the other seven patients were negative in ELISA despite showing IgE-reactivity in WB. It may be due to the different solubilization strategies for proteins in the two analyses. There can be some insoluble parts in the whole extract that are more exposed to denaturing buffers used for SDS-PAGE compared to ELISA; therefore, the epitopes are more accessible in gel than they are in ELISA. It is important to emphasize that the extract produced for electrophoresis and blotting did not aim to simulate the sensitization process but to bring as many proteins/allergens into the solution for the in vitro assays. Additionally, it should also be considered that more than one collection of pollen was made over two consecutive years in order to have enough extract to use in all experiments and the extract used in ELISA was not the same as that used for the proteomic approach. Zheko et al. [34] showed that pollen collected in consecutive years and also in different periods of the same year could present different protein content with variations even from one week to the next. Besides, the natural variation of allergens from biological sources was reported for many allergen sources, including birch pollen [35] and olive pollen [36].

In clinics, cashew tree pollen is already considered an important trigger of rhinitis symptoms in the northeast region of Brazil and can also be important in other countries that present and cultivate cashew trees such as India, Vietnam, Thailand, Indonesia, Philippines, and West and East African countries [37]. However, currently, no standardized cashew tree pollen extract is available to unambiguously confirming patients’ sensitization. Therefore, it would be ideal to have a proper extract to be used for diagnosis, especially in nasal provocations, to confirm the allergenic potential of this pollen. It is not confirmed whether the same allergens found in cashew tree pollen could also be found in cashew nut or cashew kernel, but as no patients present any symptoms to other sources, it is more likely that the allergens from pollen are specific and not present in other parts of the plant.

In summary, a comprehensive study of the protein profile of cashew tree pollen was performed, and a panel of molecules belonging to different allergenic protein families was identified. Proteomic results were further validated by a 2D immunoblot using patients’ sera. We have identified novel IgE-reactive molecules from cashew tree pollen that are homologous to allergens already identified in other pollens and several IgE-reactive proteins not yet described as allergenic that need to be further studied. To this end, the elucidation of the allergenic molecules from cashew tree pollen, as well as their immunological and structural characterization, will help to improve diagnostics and future treatment of under-evaluated allergen sources. Besides, this new knowledge offers tools for predicting epitopes and producing hypoallergenic molecules in the future.

## 4. Materials and Methods

### 4.1. Patient Selection

Based on the clinical history, 12 patients with persistent allergic rhinitis (>4 days per week and >4 weeks) characterized by the worsening of the symptoms at the time of flowering of cashew tree were selected at the Ambulatório de Alergia at Núcleo de Atenção Médica Integrada (NAMI) of Universidade de Fortaleza (UNIFOR) (Table 1). This research was approved by the Ethical Committee “Comissão de Ética em Pesquisa da Universidade de Fortaleza (UNIFOR)-Fundação Edson Queiroz” (CAAE: 49985615.0.0000.5052) and the “Comissão de Ética em Pesquisa da Faculdade de Medicina da Universidade de São Paulo” (CAAE: 49985615.0.3001.0065) approved on 28 January 2016. Informed consent was obtained from each subject, in accordance with the Helsinki Declaration.

Patients were submitted to a prick test with extracts of *Dermatophagoides farinae*, *Dermatophagoides pteronissinus*, *Blomia tropicalis*, mold (*Aspergillus fumigatus*, *Alternaria alternata*, *Penicilium notatum*, *Cladosporium herbarum*), mix grass pollen I (*Avena sativa*, *Hordeum vulgare*, *Secale celeare*, *Triticum sativum*), and mix grass pollen II (*Dactylis glomerata*, *Festuca pratensis*, *Lolium perenne*, *Phleum pratense*, *Poa pratensis*) from AsacPharma IPI (Alicante, Spain).

Five non-atopic individuals without history of rhinitis living in the same area were selected as controls.

### 4.2. Pollen Extract Preparation

Cashew (*Anacardium occidentale*) flowers were collected in Fortaleza-CE during the flowering season, from August to October in 2014 and 2015. Anthers were manually separated with tweezers and scalpel and analyzed by fluorescence microscopy Nikon ED 3 (Nikon, Melville, NY, USA) to verify the presence of pollen. After the confirmation of pollen content, grains were manually extracted with tweezers in PBS using magnifying lenses for measuring the diameters via a phase contrast microscopy EVOS AME-330 (Waltham, MA, USA) using ImageJ (version 1.52) software (National Institutes of Health, Bethesda, MD, USA).

The protein extract was produced from the maceration of pollen sacs by phenolic extraction and precipitation using 100 mM ammonium acetate/methanol in the presence of protease inhibitors adapted from Carpentier et al. [38].

The protein concentration of the extract was measured by Bradford [39], and the extract was stored at −20 °C until use. Out of different collections, three extracts were prepared, two from 2014 for the proteomic approach and IgE-reactive determination and one from 2015 for ELISA.

### 4.3. Mass Spectrometry, Protein Identification, and GO Annotations

For shotgun analyses, both pollen extracts and IgE-reactive spots from 2D gels were digested with the ProteoExtract All-in-One Trypsin Digestion Kit (Calbiochem, Gibbstown, NJ, USA). After desalting with Millipore ZipTip C18 (Merck KGaA, Darmstadt, Germany), peptides were loaded on a Acclaim PepMap RSLC column (C18, 75 µm × 15 cm, Dionex, Thermo Fisher Scientific, Waltham, MA, USA), and the column was developed with an acetonitrile gradient (Solvent A: 0.1% (*v*/*v*) formic acid; solvent B: 0.1% (*v*/*v*) formic acid/90% (*v*/*v*) acetonitrile; 5%–45% B in 60 min) at a flow rate of 300 nL/min at 55 °C. The HPLC (Dionex Ultimate 3000, Thermo Fisher Scientific, Waltham, MA, USA) was directly coupled via nanoelectrospray to a Q Exactive Orbitrap mass spectrometer (Thermo Fisher Scientific). The capillary voltage was 2 kV. For peptide identification, a top 12 method was used, with the normalized fragmentation energy at 27%.

Data analysis was performed with PEAKS Studio 7 software (Bioinformatics Solutions, Waterloo, Canada) using the NCBI protein database for plants (Viridiplantae). Cross-species searches of homologous proteins were done with the SPIDER module of PEAKS Studio. Proteins were then filtered by −10logp > 15, the exclusion of the uncharacterized and selection of those with >3 unique peptides for selection of high confidence, non-redundant proteins.

In order to describe the entire proteome of cashew tree pollen, the non-redundant list of identified proteins was annotated to gene ontology using PantherDB (www.pantherdb.org). Pie chart graphs were created for biological processes, cellular components, and molecular functions.

The identified proteins were submitted to the Allfam database [40] (http://www.meduniwien.ac.at/allfam/) to verify the existing classification into an allergenic protein family.

### 4.4. 1D and 2D Gel Electrophoresis 

For 1D gel electrophoresis, 20 µg pollen extract solubilized in denaturing/reducing sample buffer was applied to each well of 12% SDS-PAGE. In 2D electrophoresis, for the isoelectric focusing (IEF), the proteins were solubilized in a rehydration solution (DeStreak plus 0.5% IPG buffer (GE Healthcare Biosciences AB, Uppsala, Sweden) and bromophenol blue). Immobiline Dry Strips (IPG) with a pI range of 3–10 or 4–7 were rehydrated in this protein solution for 15 h, aiming at a good separation of proteins from different pIs probably encountered in the extract. The strips were submitted to IEF in an IPGphor system (GE Healthcare Biosciences AB, Uppsala, Sweden) at 300, 1000 and 5000 V until reaching 5750 Vh. For reduction and alkylation, two steps were performed, first with 1% DTT for 15 min, followed by 4% iodoacetamide (IAA) for 15 min, before the strips were loaded onto 12% polyacrylamide gels for protein separation by reducing SDS-PAGE. Proteins were stained with Coomassie Blue Colloidal [41], and gel images were captured using ImageQuant LAS 4000 (GE Healthcare Biosciences AB, Uppsala, Sweden).

### 4.5. 1D and 2D Immunoblot 

After separation, proteins of the 1D and 2D gel were electro-transferred onto nitrocellulose membranes and subsequently blocked with TBS-0.1% (*v*/*v*) Tween 20 and 5% (*w*/*v*) skimmed milk powder for 1 h at room temperature. Membranes were incubated with patients’ sera at (1:10), individually for 1D and pooled for 2D, overnight at 4 °C. Bound IgE was detected with 1:7500 goat anti-human IgE (ε), HRPO conjugate (Invitrogen-California, Carlsbad, CA, USA). Immunoreactive bands were scanned by ImageQuant LAS 4000 (GE Healthcare Biosciences AB, Uppsala, Sweden) using the Enhanced Chemiluminescence Kit (ECL) (GE Healthcare Biosciences, Uppsala, Sweden).

### 4.6. ELISA (Enzyme-Linked Immunosorbent Assay)

For ELISA experiments, Costar plates (Corning Incorporated, NY, USA) were coated with 50 μL of a PBS solution containing 4 μg/mL of pollen extract or recombinantly produced Bet v 6 [42] and incubated overnight at 4 °C. Non-specific binding was blocked with 0.05% (*v*/*v*) TBS buffer, pH 7.4 Tween-20, and 1% (*m*/*v*) BSA and incubated with 1:5 diluted patients’ sera overnight at 4 °C. Specific IgE was detected with 1:7500 human anti-IgE produced in mice (ε-specific) conjugated to Horseradish Peroxidase (HRP) (Invitrogen-California, Carlsbad, CA, USA), and the reaction was developed using the p-nitrophenyl phosphate substrate (pNPP) that shows yellow light capable of being detected in the optical reader at 405 nm. The measurements were performed in duplicates, and the values of optical density that exceeded three times the background were considered positive.

### 4.7. Phylogenetic Tree Construction

Tax_ids were retrieved from Pubmed for all the species in the list of IgE-reactive proteins identified, plus *Anacardium occidentale*, excluding common names using the tool taxonomy identifier (https://www.ncbi.nlm.nih.gov/Taxonomy/TaxIdentifier /tax_identifier.cgi).

A phylogenetic tree, based on the tax_ids and species names, was built using R (version 3.4.3), in R Studio (version 1.1.423), with the package “taxize.” The species with the closest taxonomy to *Anacardium* (belonging to the order Sapindales) were highlighted in bold.

## Figures and Tables

**Figure 1 ijms-20-02397-f001:**
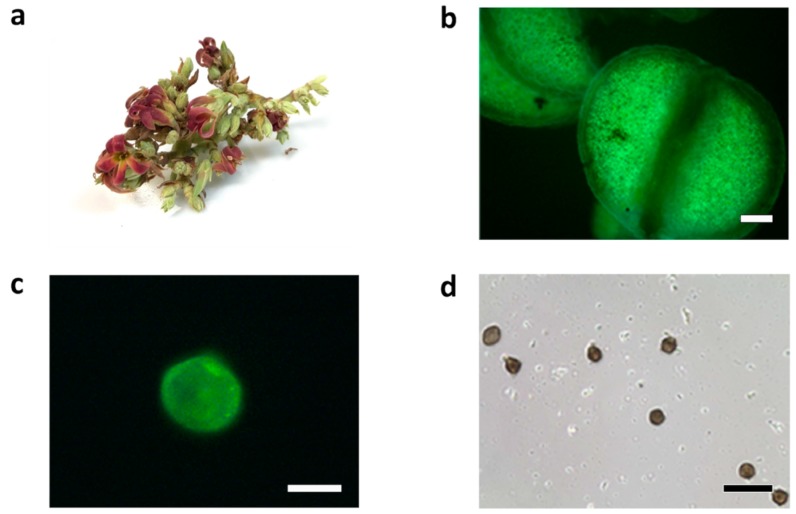
Cashew flowers and pollen particles. (**a**) a cashew flower, (**b**) autofluorescent image of anthers from a cashew flower (scale bar represents 100 µm), (**c**) autofluorescent image of pollen grains from a cashew flower (scale bar represents 20 µm), (**d**) pollen grain from a cashew flower used for measurement (scale bar represents 100 µm).

**Figure 2 ijms-20-02397-f002:**
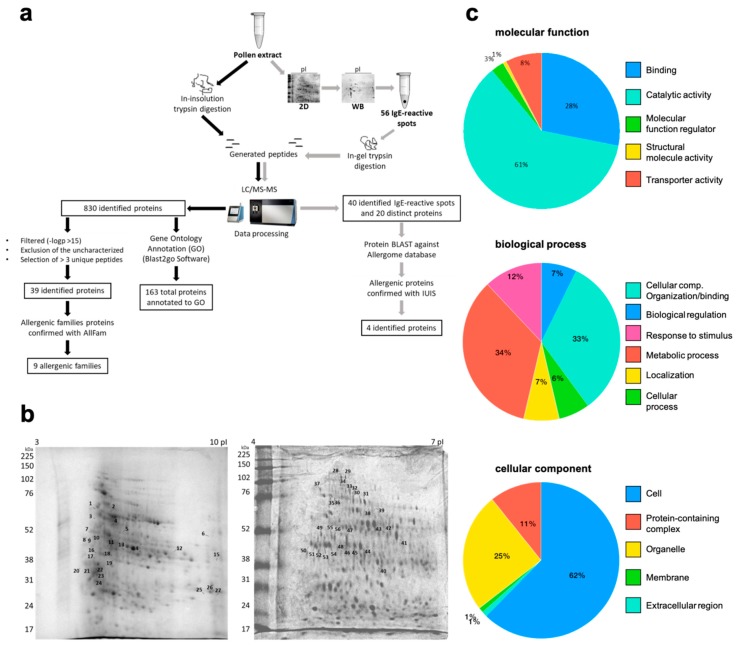
Proteomics of cashew tree pollen reveals the presence of allergenic protein families and IgE-reactive molecules. (**a**) Summarized workflow and results of the proteomic analyses. (**b**) 2D-SDS-PAGE (isoelectric point (pI) 3–10 and 4–7) stained by Coomassie. All excised spots subjected to mass analysis are numbered (1–27 and 28–56). (**c**) Gene ontology annotation of identified proteins from cashew tree pollen using PantherDB. A total of 163 proteins were annotated to a biological process, cellular component, and molecular function.

**Figure 3 ijms-20-02397-f003:**
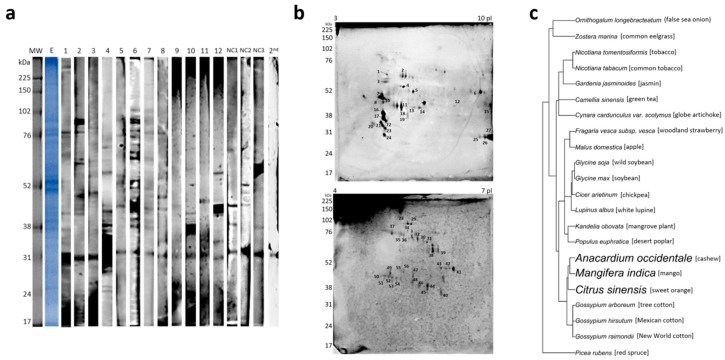
Immunoblot analysis of cashew tree pollen extract. (**a**) Pollen extract (E) on reducing SDS-PAGE and stained with Coomassie. Immunoblot of individual allergic patients’ sera (*n* = 12), negative controls (NC1, NC2, and NC3) and second antibody control (2nd). (**b**) 2D Western blotting (2D WB) showing the IgE binding of pooled sera (1–12) to cashew tree pollen extract. (**c**) Phylogenetic tree built from species found in homolog proteins identified after 2D WB. Species close to *Anacardium occidentale* are highlighted in bold. pI, isoelectric point.

**Figure 4 ijms-20-02397-f004:**
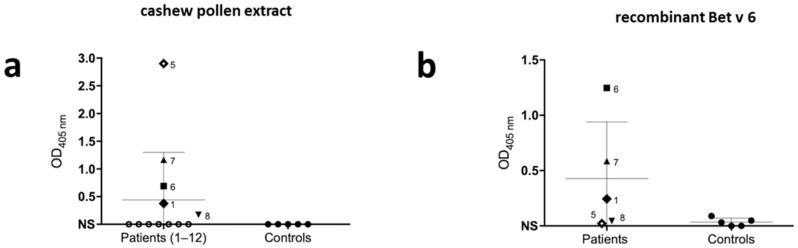
Cashew tree pollen and recombinant birch isoflavone-reductase (Bet v 6) are recognized by patients’ sera in ELISA. (**a**) IgE reactivity to immobilized cashew tree pollen extract was assessed with individual patients’ sera (*n* = 12) and non-atopic human sera (*n* = 5). (**b**) IgE reactivity to immobilized recombinant Bet v 6 was assessed with patients’ sera that were positive in ELISA for cashew tree pollen extract (*n* = 5) and non-atopic human sera (*n* = 5). The NS indicates the response threshold calculated as 3× SD of the buffer control signal.

**Table 1 ijms-20-02397-t001:** Allergenic protein families and proteins identified in cashew tree pollen by shotgun analysis.

#	Allergenic Protein Family	Protein	Organism	UP	Accession Number
1	β-1,3-glucanase and X8 domain	β-galactosidase	*Mangifera indica*	3	gi|68161828
2	Isoflavone reductase family	Isoflavone reductase	*Corylus avellana*	3	gi|532961639
3	Heat shock protein Hsp70	Heat shock 70 kDa protein	*Corchorus capsularis*	3	gi|1137176360
4	Glyceraldehyde 3-phosphate dehydrogenase	6-phosphogluconate dehydrogenase, decarboxylating	*Tarenaya spinosa*	3	gi|90657561
5	EF hand family	Calmodulin	*Actinidia valvata*	5	gi|149208297
6	Ubiquinol-cytochrome C reductase 14kD subunit	GDP-mannose 3,5-epimerase 1	*Nicotiana tabacum*	4	gi|1025343031
7	Lactate/malate-dehydrogenase	Malate dehydrogenase	*Punica granatum*	4	gi|1208527593
8	Cu/Zn superoxide dismutase	Superoxide dismutase [Cu–Zn]	*Picea sitchensis*	3	gi|116783025
			*Zoysia japonica*	4	gi|1150667722
9	Fructose-bisphosphate aldolase Class I	Fructose-bisphosphate aldolase	*Arabis alpina*	3	gi|674241776
10	-	Glutamine synthetase	*Camellia sinensis*	3	gi|67423358
			*Helianthus annuus*	3	gi|1191643447
11	-	26S protease regulatory subunit 6A	*Ananas comosus*	4	gi|1035941585
12	-	26S proteasome non-ATPase regulatory subunit 2 homolog A isoform X1	*Nelumbo nucifera*	5	gi|720059466
13	-	3-ketoacyl-CoA thiolase 2, peroxisomal-like protein	*Gossypium arboreum*	3	gi|728824325
14	-	40S ribosomal protein SA	*Daucus carota subsp. sativu*	3	gi|1021032195
15	-	Aconitate hydratase	*Lupinus angustifolius*	3	gi|1102705066
16	-	Adenosylhomocysteinase	*Daucus carota subsp. sativus*	6	gi|1040913337
17	-	ADK domain-containing protein/ADK_lid domain-containing protein (Fragment)	*Cephalotus follicularis*	3	tr|A0A1Q3BK62
18	-	Aldehyde dehydrogenase 2B7 copy 2	*Bixa orellana*	4	gi|995952955
19	-	Alpha-1,4-glucan-protein synthase [UDP-forming] 1	*Cicer arietinum*	5	gi|502148850
20	-	ATP synthase subunit alpha	*Nelumbo nucifera*	13	gi|1052487924
21	-	ATP-dependent clp protease ATP-binding subunit clpa-like cd4b, chloroplastic	*Nicotiana attenuata*	3	gi|1102148367
22	-	Band_7 domain-containing protein	*Cephalotus follicularis*	3	tr|A0A1Q3C6R4
23	-	Bifunctional aspartate aminotransferase and glutamate/aspartate-prephenate aminotransferase-like	*Nelumbo nucifera*	3	gi|720042258
24	-	Caffeoyl-CoA O-methyltransferase	*Morus notabilis*	4	gi|703143062
25	-	Citrate synthase	*Spinacia oleracea*	3	gi|902182157
			*Helianthus annuus*	4	gi|1191685410
26	-	Clathrin heavy chain	*Corchorus capsularis*	5	gi|1137169669
27	-	Elongation factor Tu	*Phaseolus vulgaris*	4	gi|561035855
28	-	Eukaryotic initiation factor 4A-15	*Ananas comosus*	4	gi|1035961582
29	-	fumarate hydratase 1, mitochondrial	*Cicer arietinum*	3	gi|502117760
30	-	Glucose-6-phosphate isomerase	*Nelumbo nucifera*	4	gi|719968717
31	-	Glutamate dehydrogenase	*Jatropha curcas*	3	gi|643706362
32	-	Ketol-acid reductoisomerase	*Nelumbo nucifera*	3	gi|720041220
33	-	NADH dehydrogenase [ubiquinone] iron-sulfur protein 1, mitochondrial	*Cicer arietinum*	4	gi|502137547
34	-	Peroxisomal (*S*)-2-hydroxy-acid oxidase GLO1-like	*Nicotiana tabacum*	3	gi|1025062071
35	-	Phosphoenolpyruvate carboxylase	*Arabidopsis lyrata subsp. lyrata*	3	gi|297327702
36	-	Purple acid phosphatase	*Vitis vinifera*	3	gi|147771668
37	-	Rhamnogalacturonate lyase family protein	*Theobroma cacao*	3	gi|508723620
38	-	Sucrose synthase	*Mangifera indica*	5	gi|425875159
39	-	Ubiquitin-conjugating enzyme E2 variant 1D	*Vigna radiata var. radiata*	5	gi|951030072

UP, unique peptide.

**Table 2 ijms-20-02397-t002:** Demographic, clinical features, and sensitivities of patients.

Patient	Sex	Age (y)	Symptoms	SPT					
				Der f	Der p	Blo t	Mold	Mix Grass Pollen I	Mix Grass Pollen II
1	F	2	AR, asthma, dermatitis	+	+	+	−	−	−
2	M	11	AR, asthma, conjunctivitis	+	+	+	−	−	−
3	F	6	AR, conjunctivitis	+	+	+	+	−	−
4	M	17	AR, asthma, conjunctivitis, dermatitis	+	+	+	−	−	−
5	M	13	AR, conjunctivitis	+	+	+	−	−	−
6	F	65	AR, asthma, conjunctivitis	+	+	+	−	−	−
7	M	13	AR, conjunctivitis	+	+	+	−	−	−
8	F	6	AR, asthma, conjunctivitis	+	+	+	−	−	−
9	F	38	AR, asthma	+	+	+	−	−	−
10	M	7	AR, asthma	+	+	+	−	−	−
11	F	34	AR, conjunctivitis	+	+	+	+	−	−
12	M	14	AR	+	+	+	−	−	−

y, year; Der f, *Dermatophagoides farinae*; Der p, *Dermatophagoides pteronyssinus*; Blo t, *Blomia tropicalis*; AR, allergic rhinitis; SPT, skin prick test; + positive (≥3 × 3 mm); − negative (<3 × 3 mm).

**Table 3 ijms-20-02397-t003:** Allergenic protein families and immunoglobulin E (IgE)-reactive proteins identified in cashew tree pollen by 2D-LC-MS/MS.

Spot #	Allergenic Protein Family	Protein	Organism	UP	Accession Number
2	β-1,3-glucanase and X8 domain	PREDICTED: β-glucosidase 44-like	*Citrus sinensis*	4	gi|568833751
22		β-1,3-glucanase	*Mangifera indica*	4	gi|87042321
28–29–31–35–36		β-galactosidase	*Mangifera indica*	9–6–4–5–6	gi|68161828
7	Prolamin superfamily	PREDICTED: α-galactosidase 3	*Fragaria vesca subsp. vesca*	2	gi|470144145
8	GDSL-hydrolase family	PREDICTED: GDSL esterase/lipase EXL3-like	*Malus domestica*	2	gi|658038118
19	Isoflavone reductase family	PREDICTED: Isoflavone reductase homolog	*Nicotiana tomentosiformis*	4	gi|697122051
30	Heat shock protein Hsp70	Heat shock 70 kDa protein	*Ornithogalum longebracteatum*	3	gi|731910761
32			*Morus notabilis*	2	gi|703159491
50			*Kandelia obovata*	4	gi|825119573
52			*Gossypium arboreum*	3	gi|728837176
46–48–53–54–56	Profilin	Actin	*Picea rubens*	4–3–4–3–4	gi|6103623
55			*Gossypium hirsutum*	3	gi|32186900
47	Fructose bisphosphate aldolase class I	Fructose-bisphosphate aldolase-like protein	*Gardenia jasminoides*	3	gi|721750686
3	-	α tubulin	*Nicotiana tabacum*	4	gi|11967906
4	-	H(+)-transporting two-sector ATPase	*Zostera marina*	4	gi|901802009
5	-	PREDICTED: Aminoacylase-1-like	*Populus euphratica*	4	gi|743913874
13	-	Glutamine synthetase	*Camellia sinensis*	2	gi|42733460
15	-	PREDICTED: Peroxisomal fatty acid β-oxidation multifunctional protein MFP2-like	*Glycine max*	3	gi|356564184
33	-	Hypothetical protein B456_009G415000	*Gossypium raimondii*	3	gi|763795406
39	-	T-complex protein 1 subunit β	*Glycine soja*	3	gi|734418328
40	-	Transketolase	*Cynara cardunculus var. scolymus*	3	gi|976910685
43	-	Polyubiquitin	*Lupinus albus*	3	gi|89114278
44	-	PREDICTED: Succinyl-CoA ligase [ADP-forming] subunit β, mitochondrial	*Cicer arietinum*	3	gi|502114889
45	-	Flavanone 3-hydroxylase 2	*Mangifera indica*	2	gi|724086306

UP, unique peptide.

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
