# Peer review of "Cashew Tree Pollen: An Unknown Source of IgE-Reactive Molecules"

_ijms, 2019, doi:10.3390/ijms20102397_

Round 1
Reviewer 1 Report
Figo and colleagues present a well-designed and conducted study on the use of proteomics to identify the allergen profile of cashew tree pollen, which is an important source of aeroallergens in Brazil. The results are interesting and the findings are appropriate and of interest to the readership of the journal. The manuscript would benefit from the correction of spelling and grammar, throughout the manuscript. Specifically, it would be important for the authors to discuss the possible presence of the same allergens in the cashew pollen, cashew nut, cashew juice and cashew oil. This will be important to assess if some allergens are specifically in pollen or in other parts of the plant. Figure 1c is far too dark and needs to be improved. Also, in Figure 1 legend, “Barr” needs to be replaced with “scale bar represents”. I am very concerned on the effect of phenol extraction followed by ammonium acetate precipitation in methanol on the protein extract’s ability to behave like the native allergens that sensitize the subjects in the environment. Thus, what one observes in the lab due to this harsh extraction method may not be the same as what happens with the natural sensitization process. The authors need to comment and discuss this in the manuscript. For example, what effect would this extraction have on the IgE binding capacity of the cashew pollen proteins in the 1D WB, 2D WB and ELISA? In Table 2, patients’ sensitivities to which allergens should be included. In Figure 3c, it would be nice to include the common names in brackets next to the scientific names. Also, in Figure 3a, what does NC1, NC2, NC3 and 2nd stand for? These need to be outlined in Figure 3 legend. In Figure 4, only 41.7% of the patients had IgE to cashew pollen , while 60% had IgE to rBet v6. Were the same patients used in both the cashew pollen and rBet v 6 ELISAs? If so, which spot is which patient? Also, although Figure 4a shows 7 control patients, the legend indicates only 5, which should be corrected. How were control patients selected? On page 9, lines 183-185, the authors indicate non-specific binding with the secondary antibody. Was a different secondary antibody used to troubleshoot this problem?
Author Response
Word File Uploaded

Reviewer 2 Report
This is an interesting and novel piece of work on allergenic proteins in cashew pollen. It adds significantly to the current level of knowledge about cashew tree pollen. It clearly defines approaches to identify novel allergens on a molecular level and to identify homologous proteins from proximate species.
Author Response
Word file Uploaded

Reviewer 3 Report
The basic message is fine, but this is obviously a very REGION issue,
I had a lot of English language issues. I listed most below. Is it possible to have a English proofreader read the paper and clean up English awkwardness?
L 70-71-72 sentence: not clear what is meant
L 80 cashew tree
Line 85 grains not commonly used
Line 86 fruits?
Line 171 cashew tree
Line 174 "but" without
Line 192-195 Unclear
Author Response
Word file Uploaded

Round 2
Reviewer 3 Report
1. The English language and grammar is not traditional American English, however, it's readability is acceptable
2. Make sure the term cashew tree pollen is used in all instances where cashew pollen was previously used